# Febrile Rash: An Early Diagnostic Clue to Infectious Illness in Travelers Returning from Thailand

**DOI:** 10.3390/reports7020045

**Published:** 2024-06-07

**Authors:** Hisham Ahmed Imad, Anastasia Putri, Ratchata Charoenwisedsil, Sakarn Charoensakulchai, Eric Caumes

**Affiliations:** 1Thai Travel Clinic, Hospital for Tropical Diseases, Faculty of Tropical Medicine, Mahidol University, Bangkok 10400, Thailand; anastasia@thaitravelclinic.com (A.P.); rachata@thaitravelclinic.com (R.C.); sakarn@thaitravelclinic.com (S.C.); 2Department of Clinical Tropical Medicine, Faculty of Tropical Medicine, Mahidol University, Bangkok 10400, Thailand; 3Center for Infectious Diseases Education and Research, Department of Viral Infections, Research Institute for Microbial Diseases, Osaka University, Osaka 565-0871, Japan; 4Centre de Diagnostic et de Thérapeutique, Hotel Dieu Hôpital, 75004 Paris, France; eric.caumes@aphp.fr

**Keywords:** febrile rash, dengue, Thailand, France

## Abstract

The eruption of a rash along with spiking fever in travelers returning from the tropics may be suspicious of arboviral diseases, and isolation prevent further transmission in non-endemic countries. The case presented here was seen at the Fever Clinic at the Hospital for Tropical Diseases in Bangkok, Thailand. The presenting complaints were fever, headache, myalgia, and a distinctive erythematous blanching rash. Despite a negative dengue NS1 test on the initial day, anti-dengue IgM and IgG were detectable on day five of illness. Dengue, a leading cause of traveler’s fever with rash, is of particular concern, especially during outbreaks like the one in Thailand in 2023, when the number of cases exceeded one hundred thousand over a nine-month period. The influx of 28 million travelers in 2023, many with naive immunity to many arboviruses, raises fear of transmission to temperate regions, including to countries like France, where *Aedes albopictus* establishment can lead to autochthonous dengue cases and clusters. Enhanced surveillance is crucial, urging the consideration of dengue as a potential diagnosis in travelers with febrile rash, even prior to lab confirmation. Immediate isolation of patients is essential to prevent autochthonous transmission, reduce outbreak risks, and avert public health crises.

A diagnosis for febrile rash in travelers returning from the tropics presents a diagnostic challenge. It should immediately point to arboviral disease to avoid any further transmission in the non-endemic country if the vector is widespread there. Additionally, consideration should be given to the recent increased accounts of febrile exanthema that do not require a vector, such as measles and mpox [1,2]. However, it is important to note that the characteristics of the rash in these cases greatly differ from what we describe further.

The diagnosis of arboviral disease may be oriented by the epidemiological background, the natural history of the disease, the clinical signs, and routine lab testing [3].

In September 2023, a 39-year-old male presented to the Fever Clinic at the Hospital for Tropical Diseases within hours of developing a fever (39 °C), headache, myalgia, and a distinctive rash, as shown in Figure 1.

Additionally, laboratory analyses indicated leukocytes at 4700/µL, neutrophils at 3619/µL, lymphocytes at 705/µL, and a platelet count of 177,000/µL. Unexpectedly, a dengue NS1 test returned negative on the initial day of illness, but, as anticipated, anti-dengue IgM and IgG were detectable by day five of illness.

This conundrum encountered with the rapid diagnostic test (RDT) at presentation may be attributed to various factors. Firstly, it is important to acknowledge that certain RDTs exhibit limited sensitivity and specificity. Additionally, their performance may be suboptimal based on the infecting serotype, leading to false-negative results even in the acute phase of disease. Furthermore, the optimal observation time for IgM is on the 3rd of day illness, and for IgG, it is on the 5th day of illness. We may further interpret the results if both IgM and IgG are positive five days into the illness, distinguishing between a secondary dengue infection and a primary one. The serial hematological profile and other investigations performed during the hospitalization workup are detailed in the Appendix A.

Dengue, a mosquito-borne *Flavivirus* with four serotypes, circulates in *Aedes* mosquitoes and viremic humans, ranking second only to malaria as the primary cause of fever in returning travelers, and can contribute to cluster outbreaks in individuals with naive immunity to dengue viruses [4,5].

It is important to highlight that the symptoms of dengue infection may share similarities with those of other prevalent *Flaviviruses* viruses in Southeast Asia, such as Zika virus, Japanese encephalitis virus, and even *Alphaviuses* like chikungunya virus [6,7,8,9]. In addition to malaria, other arthropod-borne rickettsioses, including murine typhus, scrub typhus, or Thai tick typhus, may present with overlapping symptoms [10,11,12,13]. Therefore, it is important to consider these diverse infectious agents with respect to the extent of exposure when assessing potential cases returning from Thailand.

Dengue virus infection poses a significant health threat in Thailand, particularly in Bangkok. A decade ago, in 2013, the incidence rates were alarmingly high, with 136.6 cases per 100,000 population [14]. In the same year, dengue infection accounted for 40% of the primary causes of illness in patients seeking treatment at the Hospital for Tropical Diseases [13]. The incidence of dengue in 2023 crossed 181.29 cases per 100,000 population according to the Ministry of Public Health in Thailand (Figure 2).

This recent large outbreak in Thailand raises concerns about potential spread to non-endemic areas (Figure 2).

The influx of over 28 million travelers in 2023, many without immunity, heightens concerns about its transmission to temperate regions where potential vectors may exist.

This is well illustrated in France, a country where *Aedes albopictus* is present. Between 1 May 2023 and 29 September 2023, a total of 1099 cases of dengue were imported to France, and Thailand emerged as the primary source of dengue-infected travelers after French overseas territories. In additional data up to 8 December 2023, a total of 2019 cases of dengue fever were observed in France, and Thailand ranked as the fourth country for most imported cases (n = 94) after Guadeloupe (612) and Martinique (755) in the French West Indies and, interestingly, one hundred cases from Mexico. Several of these imported cases led to eight autochthonous dengue clusters involving forty-three cases, with two to eleven per cluster [15]. Lastly, Thailand also ranked the first for importation of Zika cases in France during the same period.

This report emphasizes enhanced surveillance and the urgency of considering dengue as a potential diagnosis when encountering travelers with febrile rash even before confirmation with lab tests. Immediate steps must be taken to isolate the patients, preventing further autochthonous transmission. These proactive measures are essential for reducing outbreak risk and averting public health crises.

We acknowledge certain limitations to this report. As this study is retrospective in nature, we were unable to conduct molecular and serological assays to establish a confirmatory diagnosis for the presented case, subsequently limiting our ability to identify the infecting serotype. Without these limitations, this study would have been more robust. Furthermore, we cannot clearly state if this is indeed secondary dengue infection without observing the increased antibody titers from the acute phase to convalescence or using other techniques like neutralization assays.


**Materials and Methods**


De-identified data from the patient’s medical chart were retrospectively reviewed to extract clinical and laboratory information spanning from presentation to hospital discharge. Serological assays were employed to confirm or exclude common circulating arboviruses. Dengue antigen testing utilized a commercially available lateral flow kit detecting the NS1 antigen (Biosynex, Fribourg, Switzerland). Additionally, two prototype lateral-flow immunochromatography rapid point-of-test kits were used to identify chikungunya virus envelope protein 1 (E1) antigen and Zika virus non-structural protein NS1. For dengue and chikungunya serology, commercially available kits (SD, Bioline, St Ingbert, Germany for anti-dengue IgM and IgG; SD Biosensor, Inc. Gyeonggi-do, Korea for anti-chikungunya IgM and IgG) were employed in accordance with the manufacturer’s protocol.

## Figures and Tables

**Figure 1 reports-07-00045-f001:**
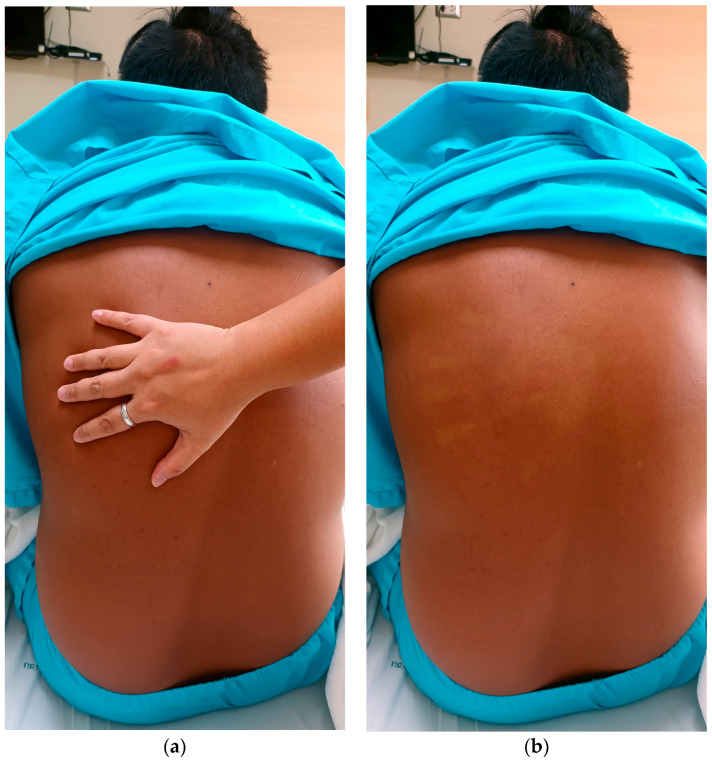
Febrile rash in dengue. (**a**) Demonstrating maneuver to exhibit blanching of acute erythematous rash in dengue. This technique exhibits the blanching effect on an acute erythematous rash in dengue. When gentle pressure is applied, the redness temporarily fades, indicating the rash’s responsiveness to pressure. (**b**) Confluent erythema covering entire back area; the distinctive feature is the clear demarcation observed on the palm of the co-author’s hand when gently pressed against the erythematous skin surface. The blanching effect observed is believed to arise from the dilation of the capillaries, a consequence of the interplay between the virus and the host response during a dengue infection.

**Figure 2 reports-07-00045-f002:**
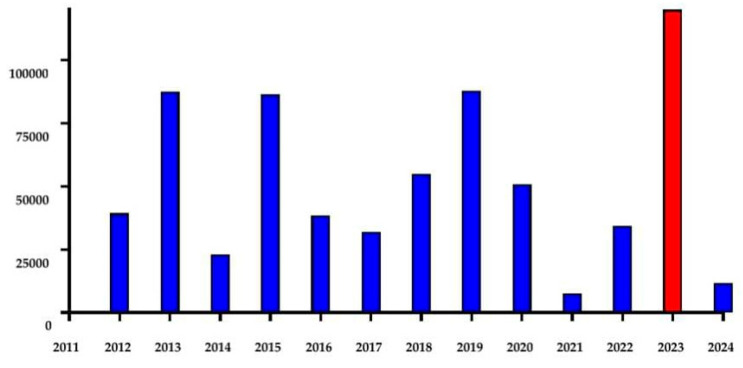
Annual incidence of dengue reported to Department of Disease Control in Thailand. The dengue outbreak in 2023 is highlighted in red.

## Data Availability

The data presented in this study are available on request from the corresponding author. The data are not publicly available to ensure the privacy of the study participant.

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
