# Peer review of "Febrile Rash: An Early Diagnostic Clue to Infectious Illness in Travelers Returning from Thailand"

_reports, 2024, doi:10.3390/reports7020045_

Round 1
Reviewer 1 Report
Comments and Suggestions for Authors
I am writing this review in my capacity as an Infectious Disease Specialist with extensive experience in the field. Having thoroughly evaluated the manuscript, I must assert that while the study is of paramount clinical importance, addressing a significant diagnostic challenge in travelers returning from tropical regions with arboviral diseases, it requires comprehensive revisions before it can be considered for publication.
Major Concerns:
1. Quality of Images: A critical aspect that requires urgent improvement is the quality of the images presented within the manuscript. Figures 1, intended to illustrate clinical manifestations and epidemiological data respectively, are not of sufficient quality for academic dissemination. Better images are essential for the effective communication of the rash characteristics critical to the diagnosis of arboviral infections. I recommend the authors to procure and provide higher quality, more detailed images that can significantly enhance the understanding of the clinical aspects discussed.
2. Analytical Depth and Discussion: The manuscript would greatly benefit from a deeper analysis and discussion of the differential diagnosis of febrile rash in travelers, the implications of early diagnosis, and the potential for preventing transmission. The current discussion, while informative, could be expanded to include a broader range of arboviruses, incorporating more recent studies and meta-analyses to provide a more comprehensive overview of the subject matter.
3. Methodological Details: The methodology section requires elaboration to clarify the diagnostic criteria, patient selection process, and the specifics of the laboratory testing employed. Such details are crucial for the reproducibility of the study and for the reader's understanding of how conclusions were drawn.
4. References and Currentness: While the manuscript cites relevant literature, there is a noticeable absence of the latest research articles on the subject. Incorporating more recent references could enhance the manuscript's relevance and provide a current perspective on the diagnosis and management of febrile rash in the context of global travel and arboviral disease transmission.
Minor Concerns
Editing and Formatting: There are several instances where the manuscript could benefit from proofreading to correct minor typographical errors and improve the overall flow of the text.
Data Presentation: The presentation of data, particularly in the results section, could be made more impactful through the use of additional charts, graphs, and tables to complement the narrative.
Conclusion:
The manuscript "Febrile Rash: An Early Diagnostic Clue in Ill-Returning Travelers from the Tropical Region" tackles a clinically significant issue with direct implications for public health, especially considering the global increase in travel and the consequent risk of arboviral diseases spreading to non-endemic regions. However, to fully realize its potential and ensure its suitability for publication, the manuscript necessitates thorough revision.
As an expert in the field, I eagerly anticipate the revised version of this article, confident that the necessary amendments will significantly elevate its contribution to the scientific community and clinical practice
Comments on the Quality of English LanguageModerate editing of language required
Author Response
Thank you very much for taking the time and reviewing our manuscript. We appreciate it a lot and welcome the suggestions recommended to improve the manuscript.
Reviewer 1 comments:
I am writing this review in my capacity as an Infectious Disease Specialist with extensive experience in the field. Having thoroughly evaluated the manuscript, I must assert that while the study is of paramount clinical importance, addressing a significant diagnostic challenge in travelers returning from tropical regions with arboviral diseases, it requires comprehensive revisions before it can be considered for publication.
Major Concerns:
- Quality of Images: A critical aspect that requires urgent improvement is the quality of the images presented within the manuscript. Figures 1, intended to illustrate clinical manifestations and epidemiological data respectively, are not of sufficient quality for academic dissemination. Better images are essential for the effective communication of the rash characteristics critical to the diagnosis of arboviral infections. I recommend the authors to procure and provide higher quality, more detailed images that can significantly enhance the understanding of the clinical aspects discussed.
Unfortunately, due to the retrospective nature of our report, the presented images are the only ones available. We acknowledge this limitation and have made efforts to enhance clarity in the footnotes by describing the depicted rash in Figure 1. We hope this additional information addresses your concern.
- Analytical Depth and Discussion: The manuscript would greatly benefit from a deeper analysis and discussion of the differential diagnosis of febrile rash in travelers, the implications of early diagnosis, and the potential for preventing transmission. The current discussion, while informative, could be expanded to include a broader range of arboviruses, incorporating more recent studies and meta-analyses to provide a more comprehensive overview of the subject matter.
As suggested, we have expanded the discussion from dengue and included other arboviruses such as Zika as evidence of other imported arboviruses to France from Thailand.
- Methodological Details: The methodology section requires elaboration to clarify the diagnostic criteria, patient selection process, and the specifics of the laboratory testing employed. Such details are crucial for the reproducibility of the study and for the reader's understanding of how conclusions were drawn.
As suggested, we have added a Materials and Methods section to the manuscript.
Materials and Methods
De-identified data from medical chart was retrospectively reviewed to extract clinical and laboratory information spanning from presentation to hospital discharge. Serological assays were employed to confirm or exclude common circulating arboviruses. Dengue antigen testing utilized a commercially available lateral flow kit detecting the NS1 antigen (Biosynex, Fribourg, Switzerland). Additionally, two prototype lateral-flow immunochromatography rapid point-of-test kits were used to identify chikungunya virus envelope protein 1 (E1) antigen and Zika virus non-structural protein NS1. For dengue and chikungunya serology, commercially available kits (SD, Bioline, St Ingbert, Germany for anti-dengue IgM and IgG, and SD Biosensor, Inc. Gyeonggi-do, Korea for anti-chikungunya IgM and IgG) were employed in accordance with the manufacturer's protocol.
- References and Currentness: While the manuscript cites relevant literature, there is a noticeable absence of the latest research articles on the subject. Incorporating more recent references could enhance the manuscript's relevance and provide a current perspective on the diagnosis and management of febrile rash in the context of global travel and arboviral disease transmission.
As suggested, we have revised the references.
Minor Concerns
Editing and Formatting: There are several instances where the manuscript could benefit from proofreading to correct minor typographical errors and improve the overall flow of the text.
Data Presentation: The presentation of data, particularly in the results section, could be made more impactful through the use of additional charts, graphs, and tables to complement the narrative.
We acknowledge the need for proof reading and have addressed all typographical errors in the revised manuscript.
Conclusion:
The manuscript "Febrile Rash: An Early Diagnostic Clue in Ill-Returning Travelers from the Tropical Region" tackles a clinically significant issue with direct implications for public health, especially considering the global increase in travel and the consequent risk of arboviral diseases spreading to non-endemic regions. However, to fully realize its potential and ensure its suitability for publication, the manuscript necessitates thorough revision.
As an expert in the field, I eagerly anticipate the revised version of this article, confident that the necessary amendments will significantly elevate its contribution to the scientific community and clinical practice
Reviewer 2 Report
Comments and Suggestions for Authors
1. The title is better to modify. Febrile Rash: An early Diagnostic Clue to the illness.............
2. The Traveler is from France? Returning from Thailand, the patient showed symptom? Dengue test done in Hospital in Thailand? Writing is not clear. Please clarify it.
3. What kind of test kit you used for dengue confirmation? ICT? Describe kit name.
4. Although 5 days after fever, NS1 was negative and DENV IgM and IgG were detected. It is still acute state and NS1 should be detect. Discuss it .
5. Did you check ZIKV infection? Rash is common in arbovirus infection (DENV, CHIKV, ZIKV). In 2023, ZIKV infection was reported in Thailand (WHO, Aug 2023) .https://www.nature.com/articles/s41598-023-48508-4.
6. Did you check the patient has primary or secondary DENV infection? Even 5 days post infection, IgG produced . Is it secondary infection?
7. Please add reference in Line 53-57. "Between May1, 2023.... per cluster".
8. In Figure, you added annual incidence of reported dengue cases. However, 2023 not included. Please update it.
Author Response
Thank you very much for taking the time and reviewing our manuscript. We appreciate it a lot and welcome the suggestions recommended to improve the manuscript.
Reviewer 2 comments:
- The title is better to modify. Febrile Rash: An early Diagnostic Clue to the illness.............
We appreciate this suggestion and have revised the title in the revised manuscript.
“Febrile Rash: an early diagnostic clue to the infectious illness in ill-returning travelers from Thailand”
- The Traveler is from France? Returning from Thailand, the patient showed symptom? Dengue test done in Hospital in Thailand? Writing is not clear. Please clarify it.
The traveler in question is not French.
Nevertheless, this situation provided an excellent opportunity to illustrate that when individual with dengue infection seeks medical attention very early in the course of their illness, rapid diagnostic tests may yield inconclusive results. Consequently, we must rely on clinical observations that support the diagnosis of arboviral infections like dengue.
In this specific instance, the individual presented to the Fever Clinic within less than 12 hours of the onset of symptoms, primarily fever, and exhibiting a distinct rash.
We believed it was crucial to use this case, along with epidemiological data from Thailand and France, to emphasize the importance of promptly identifying and isolating individuals who fall ill after returning from endemic regions. This proactive measure aims to prevent local transmission, a concern exemplified by repeated instances of autochthonous transmission in France.
- What kind of test kit you used for dengue confirmation? ICT? Describe kit name.
Due to the retrospective nature of this report, we were unable to proceed on with confirmatory test for Flaviviruses through virus isolation or detection of viral RNA
The presented case is compelling, and such cases always pose a diagnostic challenge. Based on our experience in Thailand, the primary cause of hospitalization for acute febrile undifferentiated illness is often attributed to Flaviviruses, with dengue viruses being the most commonly screened for when patients present to the hospital. However, it's important to note that commercial test kits available may cross-react with other circulating Flaviviruses in the region, adding complexity to the diagnostic process
- Although 5 days after fever, NS1 was negative and DENV IgM and IgG were detected. It is still acute state and NS1 should be detect. Discuss it .
In the revised manuscript, we have added an additional paragraph discussing why dengue NS1 antigen was negative at presentation in Lines 114-128
- Did you check ZIKV infection? Rash is common in arbovirus infection (DENV, CHIKV, ZIKV). In 2023, ZIKV infection was reported in Thailand (WHO, Aug 2023) .https://www.nature.com/articles/s41598-023-48508-4.
We were only able to screen for Zika virus using a prototype antigen test kit. However, there was no further confirmatory test done such as PCR or serology. Further in the revised manuscript we have discussed this limitation.
- Did you check the patient has primary or secondary DENV infection? Even 5 days post infection, IgG produced . Is it secondary infection?
In this particular case, neither IgM nor IgG antibodies were initially detectable. Upon retesting, both IgM and IgG antibodies were found to be positive. Given this seroconversion pattern, we are leaning towards interpreting it as a secondary dengue infection rather than a primary one. Additionally, it's worth noting that in secondary infections, the NS1 antigenemia tends to decline rapidly. However, we cannot definitively assert this without measuring the two- or four-fold increase in the titers of anti-dengue antibodies during both the acute and convalescent periods.
- Please add reference in Line 53-57. "Between May1, 2023.... per cluster".
As suggested, we have added a reference to the paragraph explaining the situation in France.
- In Figure, you added annual incidence of reported dengue cases. However, 2023 not included. Please update it.
In the revised manuscript, we have corrected this error.
Reviewer 3 Report
Comments and Suggestions for Authors
The case report “Febrile Rash: An Early Diagnostic Clue in Ill-Returning Travelers from the Tropical Region” has been reviewed.
With the large and growing number of travelers especially during holidays, the diagnosis of uncommon or non-endemic diseases in returning or incoming population an pose a potential source of transmission to autochthonous population and difficulties in diagnosis to the local physicians. This case underscores the need to suspect Dengue in a patient returning from a tropical country, where Dengue is endemic , yet there are other viral infections such as measles that should also warrant attention , albeit the high airborne transmissibility of measles virus….it doesn’t need any help from a vector!
This is my main concern; please include other exanthemas for the differential diagnosis at arrival in the introduction, first paragraph
11
Abstract:
Line 17-18 : Something is missing in this sentence
Dengue, a major cause of traveler’s fever with rash is of particular concern due to outbreak such as Thailand’s, exceeding to over one hundred thousand cases over nine months.
Line 20-21 : Not really need to state this in the abstract
We casually observe this impact in France, where Aedes albopictus presence led to 1099 imported dengue cases from Thai-
land, resulting in six autochthonous clusters.
Change to : Importation to countries like France , where Aedes albopictus is presnt, can lead to imported dengue cases and autochthonous clusters.
Line 22: Heightened vigilance is better expressed as Enhanced surveillance , please change throughout the text
Keywords : Thailand and France as keywords?
Main text:
What is the surveillance plan for arboviral diseases in Thailand? Explain briefly along with the sentence giving incidence rates of Dengue in Thailand. There is also the example of Italy’s Chikungunya virus outbreak in 2007 due to an importation from India , (G Rezza et al. Infection with chikungunya virus in Italy: an outbreak in a temperate region . 2007 Lancet 370(9602):1840-6l. DOI: 10.1016/S0140-6736(07)61779-6
Again , as I said ,not only Dengue is to be mentioned because it seems as of there is none else to check for.
Figure 2. Should be improved, why a solid black line at the end?
Improvement could be achieved by a bit of bibliographical research
Comments on the Quality of English Language
Can be improved with minor editing
Author Response
Thank you very much for taking the time and reviewing our manuscript. We appreciate it a lot and welcome the suggestions recommended to improve the manuscript.
Reviewer 3 comments:
The case report “Febrile Rash: An Early Diagnostic Clue in Ill-Returning Travelers from the Tropical Region” has been reviewed.
With the large and growing number of travelers especially during holidays, the diagnosis of uncommon or non-endemic diseases in returning or incoming population an pose a potential source of transmission to autochthonous population and difficulties in diagnosis to the local physicians. This case underscores the need to suspect Dengue in a patient returning from a tropical country, where Dengue is endemic , yet there are other viral infections such as measles that should also warrant attention , albeit the high airborne transmissibility of measles virus….it doesn’t need any help from a vector!
This is my main concern; please include other exanthemas for the differential diagnosis at arrival in the introduction, first paragraph.
Thank you for the suggestion. We totally agree that several febrile exanthema can result from a numerous infectious etiological agents. Nevertheless, in this case we are focusing on febrile rash that would be suggestive of an abroviruses and the potential for local transmission via vector. However, we do agree that measles is far more infectious and a growing concern in the present day. However, with the absence of hallmark clinical findings such as cough, coryza and conjunctivitis and the characteristic mucocutaneous involvement, in addition to the appearance and distribution of rash in the presented case is not in favor of measles.
Nevertheless, as suggested we have addressed this Lines 32-36
"Additionally, consideration should be given to the recent increased accounts of febrile exanthema that do not require a vector, such as measles and mpox. However, it is important to note that the characteristic of the rash greatly differs from what we describe further"
Abstract:
Line 17-18 : Something is missing in this sentence
We have rephrased the sentence in Lines 17-18 for reader clarity in the revised manuscript
“Dengue, a leading cause of traveler’s fever with rash is of particular concern, especially during outbreaks like the one in Thailand in 2023, where the number of cases exceeded one hundred thousand over nine month period”
Line 20-21 : Not really need to state this in the abstract
We casually observe this impact in France, where Aedes albopictus presence led to 1099 imported dengue cases from Thailand, resulting in six autochthonous clusters.
Change to : Importation to countries like France , where Aedes albopictus is presnt, can lead to imported dengue cases and autochthonous clusters.
As suggested, we have rephrased the sentence in the revised manuscript.
“The influx of 28 million travelers in 2023, many with naive immunity to many arboviruses, raises fear of transmission to temperature regions. Importation to countries like France, where Aedes albopictus establishment can lead to autochthonous dengue cases and clusters”
Line 22: Heightened vigilance is better expressed as Enhanced surveillance , please change throughout the text
Thank you for this suggestion. In the revised manuscript we have made the changes as suggested.
Keywords : Thailand and France as keywords?
We considered including Thailand and France among the key words since, the importation of arboviruses from Thailand to France has been documented previously. We aim to emphasize this crucial aspect by including 'France' and 'Thailand' as keywords in our article. This deliberate choice enhances the visibility of our research in search results, drawing attention to the documented cross-border transmission of arboviruses and its implications
."Main text:
What is the surveillance plan for arboviral diseases in Thailand? Explain briefly along with the sentence giving incidence rates of Dengue in Thailand. There is also the example of Italy’s Chikungunya virus outbreak in 2007 due to an importation from India , (G Rezza et al. Infection with chikungunya virus in Italy: an outbreak in a temperate region . 2007 Lancet 370(9602):1840-6l. DOI: 10.1016/S0140-6736(07)61779-6
Again , as I said ,not only Dengue is to be mentioned because it seems as of there is none else to check for.
Figure 2. Should be improved, why a solid black line at the end?
In the revised manuscript, we have corrected this error in Figure 2
Improvement could be achieved by a bit of bibliographical research
As suggested, we have updated the reference.
Round 2
Reviewer 1 Report
Comments and Suggestions for Authors
The authors have significantly improved the manuscript. It is of high importance for physicians and scientist worldwide. It should be published as soon as possible.
Reviewer 3 Report
Comments and Suggestions for Authors
The authors have made great improvements to the article according to reviewers' specifications